# Relationship between Uranium Minerals and Pyrite and Its Genetic Significance in the Mianhuakeng Deposit, Northern Guangdong Province

Lirong Li [1,2], Zhengqi Wang [1,2,*] and Deru Xu [1,2,3,*]

1   State Key Laboratory of Nuclear Resources and Environment, East China University of Technology, Nanchang 330013, Jiangxi, China; 201800818001@ecut.edu.cn
2   School of Earth Sciences, East China University of Technology, Nanchang 330013, Jiangxi, China
3   CAS Key Laboratory of Mineral and Metallogeny, Guangzhou Institute of Geochemistry, Chinese Academy of Sciences, Guangzhou 510640, Guangdong, China
*   Correspondence: zhqwang@ecit.cn (Z.W.); xuderu@ecut.edu.cn (D.X.)

**Abstract:** Granite-related uranium ore is an important uranium resource type in China and worldwide. Whether the uranium geochemical theory "$U^{6+}$ oxidative migration and $U^{4+}$ reductive precipitation" is applicable to the granite-related uranium mineralization theory has not been determined. Detailed field and petrographic work, as well as scanning electron microscopy energy spectrum analysis, are conducted in this study to analyze the relationship between uranium minerals and pyrite from different ore types and evaluate the mechanism for the precipitation and enrichment of uranium in the Mianhuakeng uranium deposit of northern Guangdong. Uranium ore bodies in the Mianhuakeng deposit generally occur as vein-filling or vein-disseminated types. Four different kinds of ores are recognized: fluorite, carbonate, siliceous, and reddening types. Despite differences in the mineral assemblages, veined ores share similar characteristics and show that uranium minerals (1) occur in the central part or periphery of vein-filling ores or in interphase arrangements with syn-ore fluorite, quartz, or calcite veins; (2) occur as veinlets or are disseminated in cataclastic altered granite; (3) are inlaid with gangue minerals, primarily calcite, fluorite, and microcrystalline quartz; and (4) are closely associated with pyrite in aggregates or relatively independent states, forming straight boundaries with syn-ore gangue minerals that have euhedral and intact crystals and show mosaic growth features. All these results indicate that both pyrite and uranium minerals are co-crystallized products of the ore-forming fluid. Combined with previous research suggesting that the reducing fluid was sourced from mantle, this study shows that decreased pressure and temperature, as well as changes in pH and the solubility (saturation) of changes, rather than the redox reaction, caused the uranium precipitation in the Mianhuakeng deposit.

**Keywords:** granite-type uranium ore; uranium mineral; pyrite; precipitation mineralization mechanism; Mianhuakeng deposit



## 1. Introduction

Since the 1980s, breakthroughs in uranium prospecting and scientific research on granite-related and volcanic-related mineralization, theories of uranium mineralization, including magma differentiation uranium mineralization theory, hot water extraction theory (ascension theory), epithermal fluid transformation theory, and continental weathering theory (descension theory) have enabled Chinese geologists to propose dual-mixed, hot spot-related [1], and complex crust–mantle interactions [2,3]. Consequently, major breakthroughs have been made in the theory regarding granite- and volcanic-related hydrothermal uranium mineralization in China. However, the focus of these theories revolves around the sources of uranium and ore-forming fluids and does not involve in-depth analyses of the migration of uranium and precipitation mechanism of uranium from ore-forming fluids.

Based on the results of experiments under normal temperature and pressure and uranium geochemical characteristics in textbooks or writings, uranium in nature is believed to have two main valence states: $U^{4+}$ and $U^{6+}$. $U^{6+}$ is soluble, while $U^{4+}$ is insoluble. Uranium will be oxidized to $U^{6+}$ and migrate with ore-forming fluids under oxidizing conditions. When migrating to a reducing environment, uranium in the form of $U^{6+}$ in the fluids is reduced to insoluble $U^{4+}$, which then precipitates [4–8]. The "uranium oxidizing migration and reducing precipitation" theory is well accepted and well demonstrated in supergene uranium mineralization systems. However, many geological phenomena of granitic uranium deposits cannot be easily explained by the oxidizing migration and reducing precipitation theory. For example, pitchblende and pyrite coexist with gangue minerals, such as calcite, fluorite, and quartz, and fluid inclusions are enriched in reducing components, such as $H_2$, $CH_4$, $CO$, and $H_2S$, thus showing the reduced nature of the ore-forming fluid [9–12]. The characteristic "red" color of cataclastic altered rock type uranium ores has been proposed as a product of strong hematization resulting from the reduction of $U^{6+}$ in the fluid by pyrite during the mineralization stage. However, this hypothesis contradicts the finding that uranium is oxidized and migrates out of the hematite mineralization zone in the process of hypergenesis. Moreover, during granite-related uranium mineralization, the ore-forming fluid always migrates from a relatively closed and deep environment to a relatively open and shallow environment. A large amount of geochemical evidence has shown that granite-related uranium mineralization is related to mantle-derived fluids [2,3,13–22]. Therefore, whether the "uranium oxidizing migration and reducing precipitation-mineralization" theory of hydrothermal uranium mineralization can explain the evolution process of ore-forming fluids and the reduced properties of mantle-derived fluids in granite-related deposits is controversial.

In addition, the study of uranium migration-precipitation mechanisms in granite systems is important, especially in relation to ore prospecting and deposit evaluation. Thus, this topic needs to be studied in depth. By taking the Mianhuakeng deposit in northern Guangdong as an example, the occurrence style and coexisting relationship between uranium minerals and pyrite are investigated. The paper also discusses precipitation-mineralization mechanisms during the formation of this granite-related uranium deposit with the aim of deepening the understanding of granite-related uranium mineralization theory.

## 2. Geological Background

The Changjiang uranium ore field is located in the Zhuguang batholith, which is situated between the southwest margin of the Fujian-Jiangxi post-Caledonian uplift and the southeastern margin of the Hunan-Guangxi-Guangdong Hercynian-Indosinian depression belt in the northern South China Block. The Zhuguang batholith is a giant multistage granite complex that intrudes into Cambrian and Devonian strata and spreads in an east-west direction with frequent magmatic activity. It is mainly composed of Caledonian, Hercynian, Indosinian, and Early Yanshanian granites, among which the Indosinian and Yanshanian granites make up the main components. Regionally, the Cretaceous faulted Red basin (Nanxiong Basin) is developed in the southeastern Zhuguang batholith.

The Changjiang uranium ore field is an important part of the granite-related uranium metallogenic belt in Nanling and represents a typical example in the Zhuguang uranium mineralization zone. Indosinian medium- to fine-grained porphyritic two-mica granite ($\gamma_5^{1-3}$) (Figure 1) dated at $232 \pm 4$ Ma (zircon age SHRIMP U-Pb; [23]) is well developed in the middle and eastern areas of the Changjiang uranium ore field, whereas Early Yanshanian medium- to coarse-grained porphyritic two-mica granites ($\gamma_5^{2-1}$) and biotite granite ($\gamma_5^{2-2}$) (Figure 2) with intrusion ages between 164 and 155 Ma are the main components of the western and deep parts. In addition, both periods of granites were later intruded by NE-, NW-, and nearly E-W-trending mafic dikes (110−90 Ma; [23]) and a small amount of syenite.

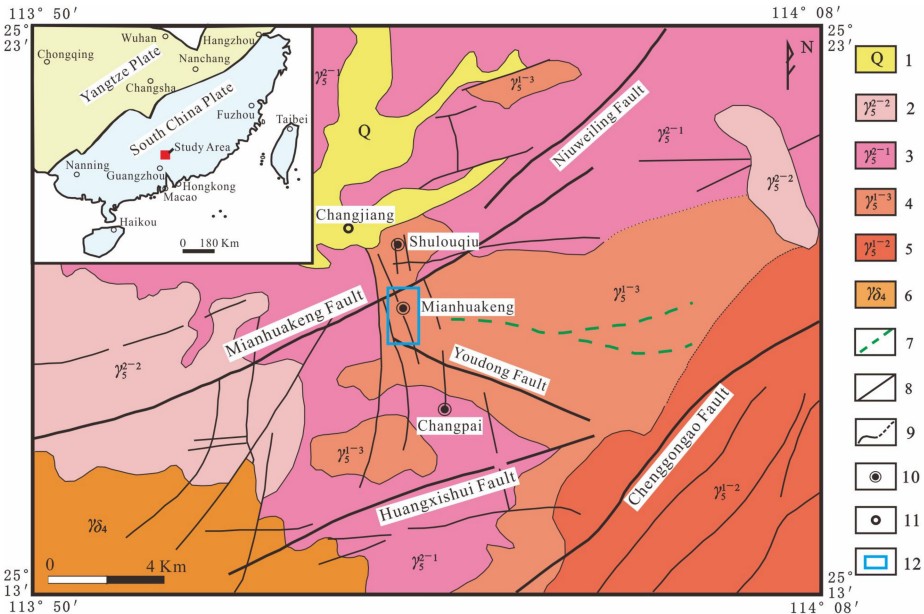

**Figure 1.** Sketch geological map of the Changjiang uranium area (modified from Huang et al. 2014 [23]). 1—Quaternary; 2—Medium-grained (porphyritic) biotite-muscovite granite; 3—Medium–coarse-grained porphyritic biotite monzonitic granite; 4—Medium–fine-grained biotite-muscovite granite; 5—Medium-grained (porphyritic) biotite granite; 6—Granodiorite; 7—Diabase; 8—Fault; 9—Measured and inferred geological boundaries; 10—Uranium deposit; 11—Place mark; and 12—Mianhuakeng deposit area.

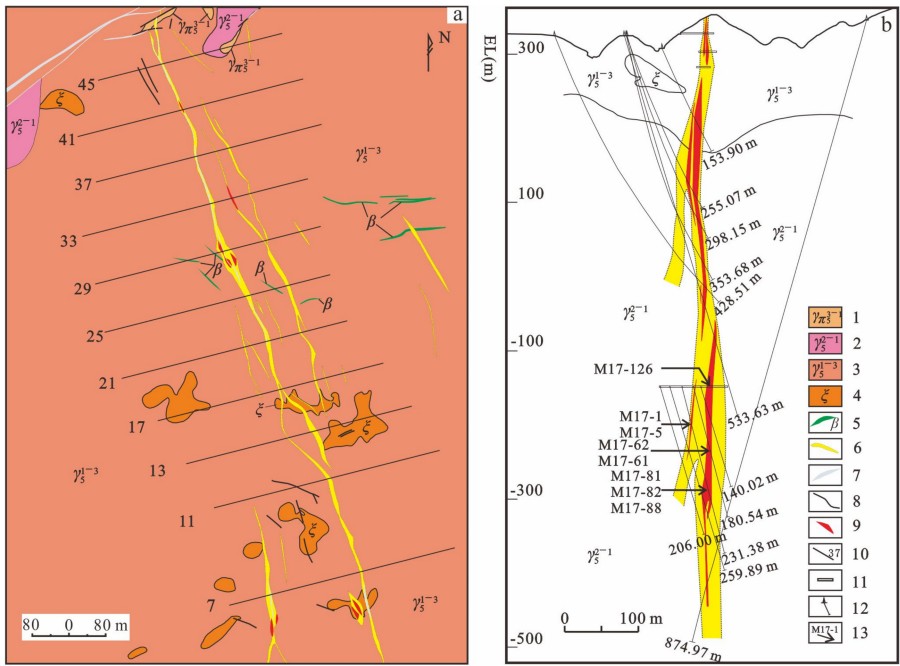

**Figure 2.** Sketch geological map (**a**) and exploration section (**b**) of the Mianhuakeng deposit. 1—Granite porphyry; 2—Medium-coarse-grained porphyritic biotite monzonitic granite; 3—Medium-fine-grained biotite-muscovite granite; 4—Syenite; 5—Diabase; 6—Structure alteration zone; 7—Silicified belt or Siliceous veins; 8—Fault; 9—Uranium ore body; 10—Exploration line and number; 11—Underground tunnel; 12—Drill hole; and 13—Sample site.

The region has experienced intense and complicated fault structural episodes and has mainly developed four groups of faults, which are NE-, NNE-, NW-, and near SN-trending and characterized by multiple movements. Among them, the large-scale NE-striking Mianhuakeng faults that are expressed in the form of large quartz veins spread throughout the whole region constitute the principal tectonic framework. Near SN-tre nding faults are the main ore-bearing structures in the orefield. By contrast, they exhibit relatively smaller scales and extension lengths (3–10 km) and form silicification belts (or veins) with different scales. Green alteration belts with widths of 10–50 m usually develop along both sides of the silicification belt, and the alteration types mainly include illitization, chloritization, silicification, redatization, carbonatization, fluoridation, pyritization, etc. Three uranium deposits, Changpai, Mianhuakeng, and Shulouqiu, have been explored in the Changjiang uranium ore field (Figure 1). Spatially, the ore bodies of the Shulouqiu deposit are distributed north of the Mianhuakeng fault, whereas the Mianhuakeng and Changpai deposits are located south of this fault, and they are approximately 4 km apart. The host rock and its alteration type, ore body space distribution, ore type, and ore-forming age of the three uranium deposits are basically similar. In particular, the Mianhuakeng deposit is one of the largest granite-related uranium deposits in South China and presents deeply buried features and a large vertical range of mineralization. As a result of its representativeness of metallogenic characteristics, the Mianhuakeng deposit has become a focus of uranium metallogenic research.

## 3. Ore Body Geology and Uranium Ore Types

The Mianhuakeng uranium deposit is located near the SN-trending faulted alteration zone of the Changjiang uranium ore field (Figure 2a). The uranium ore bodies are mostly nearly vertical concealed veins and lenses and range from depths of +500 m to −647 m. The host rocks include Indosinian medium- to fine-grained porphyritic two-mica granite and Early Yanshanian medium- to coarse-grained porphyritic two-mica granite (Figure 2b). In addition to reddening, silicification, fluoritization, carbonation, and pyritization are prominent alteration styles adjacent to the ore body. Furthermore, a much larger "greenish alteration zone" is distributed around the ore bodies, and its occurrence form is similar to that of ore-bearing structures. This greenish alteration zone is characterized by illitization and chloritization that resulted in a color change in the surrounding granites. Moreover, additional pyrites precipitated during this alteration phase, and the alteration zone is also characterized by the symmetrical distribution of alteration intensity that horizontally decreases from the center (i.e., ore bodies) to the edge. Previous results have established a uranium mineralization age of approximately 75–65 Ma for the Mianhuakeng uranium deposit [20,24,25].

The Mianhuakeng uranium ores can be basically divided into two types according to the ore structure and typical gangue mineral component: vein-filling type and cataclastic altered rock type. The former can be further classified as carbonate (Figure 3A), fluorite (Figure 4A), and silicified vein (zone) (Figure 5A) subtypes, which can be distinguished by their representative gangue minerals such as calcite, fluorite, and chalcedony or microcrystalline quartz that formed during their respective mineralization. Mineralization usually occurs in the form of vein-like or stockwork-filling ore bodies and shows vein-type and brecciated structures. The three subtypes of ores generally show overlapping or transitional characteristics of "integrated positioning" in ore bodies. In contrast, cataclastic altered rock type ores (also known as reddening type) (Figure 6A) show characteristics of dark red or dark brown-red color and cataclastic or brecciated structures. The original granite structure of this type of ore can still be recognized. These ores are symmetrically distributed on both sides of the vein-filling type of ores.

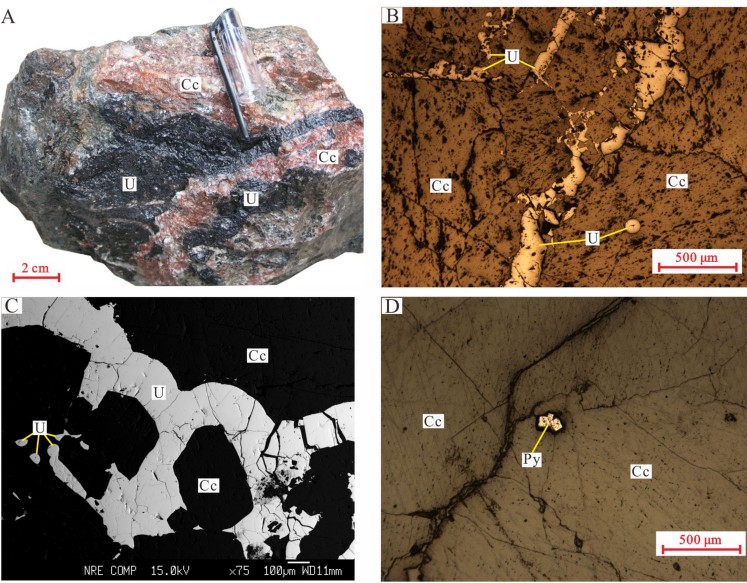

**Figure 3.** Vein uranium ore with calcite (Cc) as the main gangue mineral in vein-type uranium ores. (**A**): Pitchblende (U) occurs in the center or periphery of veins and forms a banded structure with carbonate minerals. (**B**,**C**): Pitchblende occurs in or between calcite crystals, and the crystal boundaries between pitchblende and calcite are clear and straight. (**D**): Pyrite aggregates (Py) with good crystalline morphology are encapsulated in calcites. (**A**) is the sample of uranium ore. (**B**): 5×, Reflected-light; (**C**): SEM image; and (**D**): 5×, Reflected-light. Sample No. M17-126.

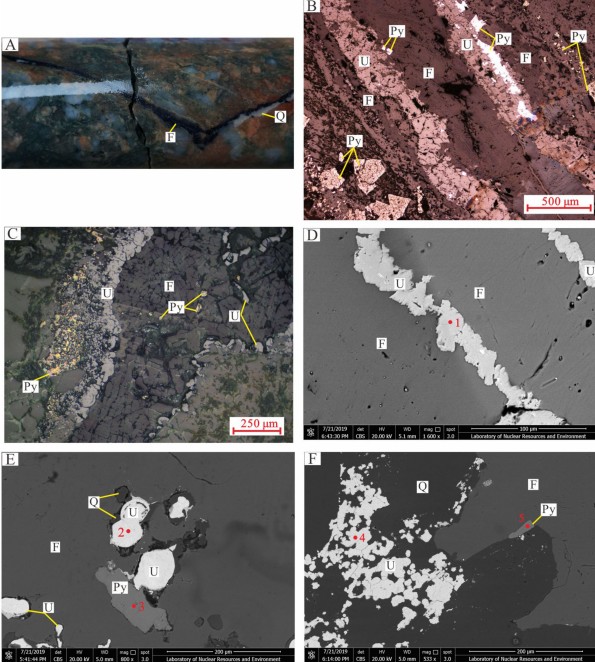

**Figure 4.** Vein uranium ore with fluorite as the main gangue mineral. (**A**): Purple-black fluorite-type uranium ore veinlets coexisting with quartz. (**B**,**C**): Center is mainly fluorite (F), with pitchblende (U), fluorite (F), and pyrite (Py) appearing successively on both sides. Pitchblende occurs in the periphery of veins; pyrite (Py) with good crystalline morphology is mainly distributed outside the vein, which is consistent with the distribution trend of vein bodies; and pitchblende and pyrite grow between fluorite grains in the vein body (**C**). (**D**): Pitchblende (U) and fluorite (F) growing alternately in belts. (**E**): Pitchblende (U) and pyrite (Py) co-crystallize between fluorite (F) crystals, and the pitchblende is surrounded by a microcrystalline quartz (Q) film. (**F**): Pitchblende (U) and pyrite (Py) growing between fluorite and quartz crystals. (**B**): 5×, Reflected-light; (**C**): 10×, Reflected-light; (**D**–**F**): SEM images. (**A**,**B**,**D**,**E**) Sample No. M17-1, and (**C**,**F**) sample No. M17-62. Graph surface digital representation energy-dispersive spectrum measuring point number.

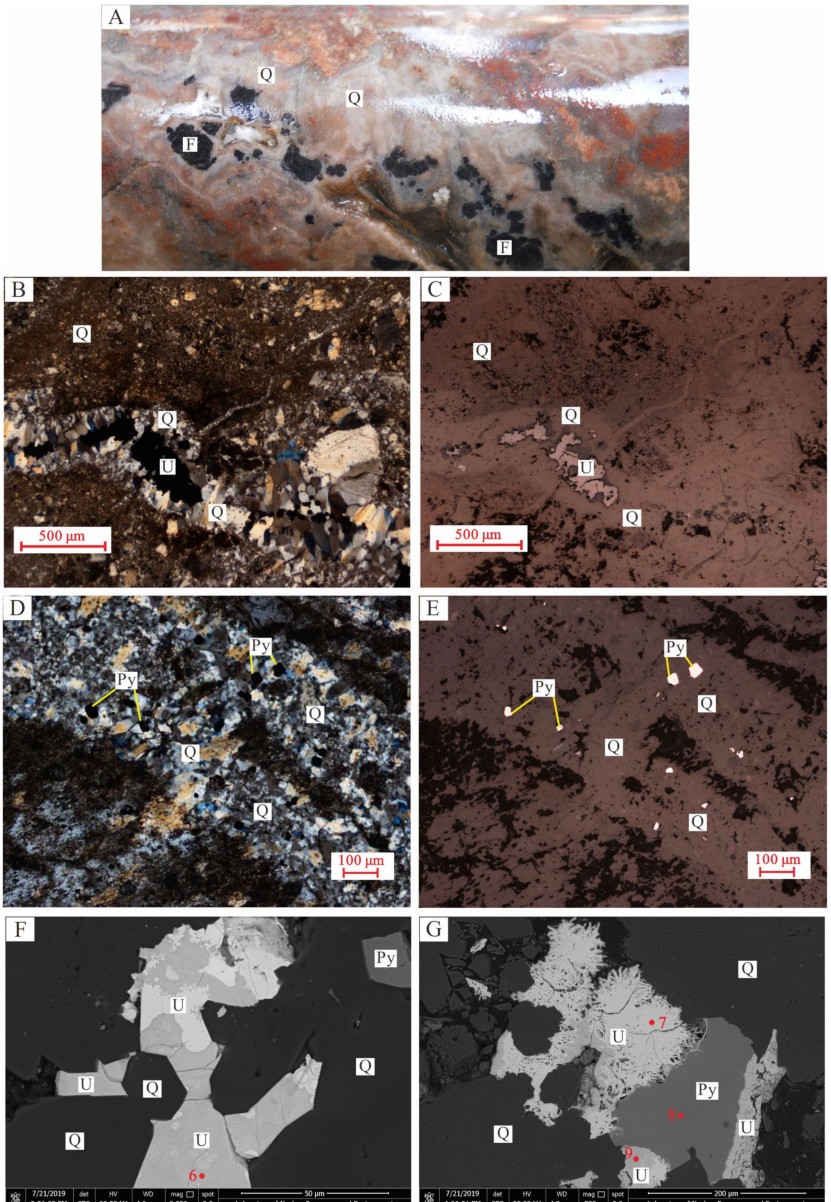

**Figure 5.** Veined uranium ore with microcrystalline quartz as the main gangue mineral. (**A**): Microcrystalline quartz and fluorite were filled between reddish clastic grains. (**B**,**C**): Pitchblende (U) occurs in the middle of the microcrystalline comb-like quartz (Q) vein. (**D**,**E**): Pyrite (Py) with a good crystal morphology embedded between quartz (Q) crystals. (**F**,**G**): Pitchblende (U) and pyrite (Py) are embedded in microcrystalline quartz (Q), and the boundaries between pitchblende, pyrite, and quartz crystals are clear and straight. (**B**): 5×, Cross-polarized light; (**C**): 5×, Reflected-light; (**D**): 20×, Cross-polarized light; (**E**): 20x, Reflected-light; (**F**,**G**): SEM images. (**A**,**B**,**C**,**F**) Sample No. M17-82, and (**D**,**E**,**G**) sample No. M17-88. Graph surface digital representation energy-dispersive spectrum measuring point number.

A gradational relationship is observed between the outer side of cataclastic altered rock ores and greenish altered wallrocks. Combined with the symmetrical distribution of alteration intensities gradually from the ore body center to both sides, it can be inferred that the ore-forming fluid filled along the fractures from a deep region and then gradually infiltrated and metasomatized the peripheral cataclastic granite.

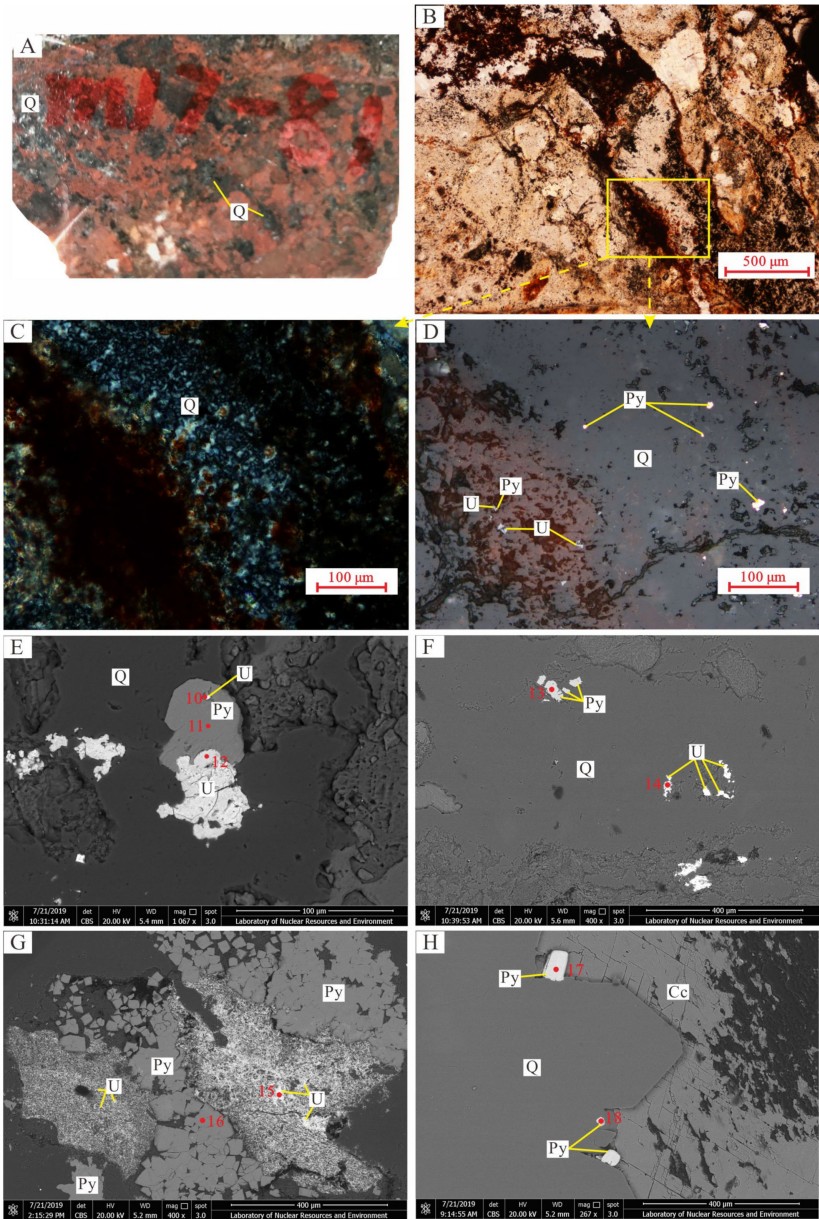

**Figure 6.** Cataclastic altered granite-related uranium ore. (**A**): Rock breccia is deep red and filled with microcrystalline quartz. (**B**): Chalcedony microcrystalline quartz veins filled between breccias. (**C,D**): Chalcedony microcrystalline quartz veins are mainly composed of microcrystalline quartz (Q), powdered rock debris, pyrite (Py), and uranium mineral (U). The powdered rock debris around uranium minerals presents a "red halo" phenomenon. Pyrite (Py) with good crystalline morphology develops an internal "red halo" between microcrystalline quartz grains. (**E,F**): Pitchblende and pyrite are embedded or independently distributed in microcrystalline quartz veins. (**G**): Uranite (U) and pyrite (Py) aggregates grow alternately in the fracture. (**H**): Pyrite (Py) with good crystal morphology crystallized between quartz (Q) and calcite minerals (Cc) in the mineralization stage. (**B**): 5×, Single-polarized light; (**C**): 20×, Cross-polarized light; (**D**): 20×, Reflected-light. (**E–H**): SEM images. (**A–F**) sample No. M17-81; (**G**) sample No. M17-61; and (**H**) sample No. M17-5. Graph surface digital representation energy dispersive spectrum measuring point number.

## 4. Genetic Relationship between Uranium Minerals and Pyrites

The samples were collected from deep boreholes or tunnels of the Mianhuakeng deposit. The specific locations are shown in Table 1 and Figure 2b. The characteristics of different ore samples are listed in Table 1. Based on macroscopic and microscopic observations combined with scanning electron microscopy (SEM) results, a detailed study

on the genetic relationship of uranium minerals and pyrites from different types of ores in the Mianhuakeng deposit is described herein. The results of the energy-dispersive spectrum analysis are listed in Table 2, while the corresponding micrographs and backscatter electron (BSE) images are shown in Figures 3–6.

**Table 1.** Uranium ore sample characteristics and location in the Mianhuakeng deposit.

| Sample No. | Ore Type | Ore Characteristic Description | Sample Location |
|---|---|---|---|
| M17-126 | Carbonate | Veined filling; center is pitchblendes, and periphery is red calcite. Figure 3A | −150 m middle ore body |
| M17-1 | | Veined filling; gangue minerals are mainly purple black fluorite, including microcrystalline quartz and calcite; and metallic minerals include uranium minerals, pyrite, etc. Reddening type uranium ore in the periphery of veins. Figure 4A. | KZK41-3, level −90.5 m |
| M17-62 | Fluorite | | KZK41-2, level −230.5 m |
| M17-82 | | Deep red; occurring as veined filling or silicified cataclastic rocks; gangue minerals are mainly microcrystalline quartz, including calcite and fluorite; and uranium minerals are symbiotic with pyrite. Reddening type uranium ore in the periphery of veins. Figure 5A. | KZK41-3, level −296.0 m |
| M17-88 | Siliceous vein | | KZK41-3, level −301.5 m |
| M17-5 | | Deep red or deep brownish red; host rock is cataclastic granite; occurring in the periphery of vein filled ore; forming minerals, such as hydromica, carbonate, pyrite, etc., and alterations, such as silicification and fluoritization. Figure 6A. | KZK41-3, level −192.5 m |
| M17-61 | Reddening | | KZK41-2, level −233.5 m |
| M17-81 | | | KZK41-3, level −289.6 m |

**Table 2.** SEM energy spectrum analytical results of mineral chemical compositions (%).

| Spot | Sample | U | O | Si | Ca | S | Fe | Al | As | Total | Minerals |
|---|---|---|---|---|---|---|---|---|---|---|---|
| 1 | M17-1 | 75.58 | 17.78 | 1.54 | 5.10 | | | | | 100 | Pitchblende |
| 2 | M17-1 | 72.49 | 20.27 | 2.34 | 4.90 | | | | | 100 | Pitchblende |
| 3 | M17-1 | | | | | 54.52 | 45.48 | | | 100 | Pyrite |
| 4 | M17-62 | 73.73 | 18.81 | 2.05 | 5.42 | | | | | 100 | Pitchblende |
| 5 | M17-62 | | | | | 52.99 | 45.79 | | 1.22 | 100 | Pyrite |
| 6 | M17-82 | 65.03 | 24.87 | 7.97 | 1.53 | | | 0.61 | | 100 | Pitchblende |
| 7 | M17-88 | 73.38 | 20.22 | 2.53 | 3.87 | | | | | 100 | Pitchblende |
| 8 | M17-88 | | | | | 53.03 | 46.97 | | | 100 | Pyrite |
| 9 | M17-88 | 62.87 | 25.64 | 9.01 | 1.76 | | | 0.72 | | 100 | Pitchblende |
| 10 | M17-81 | 65.22 | 23.43 | 3.72 | 2.05 | 2.13 | 2.78 | 0.68 | | 100 | Pitchblende |
| 11 | M17-81 | | | | | 54.04 | 45.96 | | | 100 | Pyrite |
| 12 | M17-81 | 62.96 | 27.89 | 7.46 | 0.97 | | | 0.72 | | 100 | Pitchblende |
| 13 | M17-81 | | | | | 53.97 | 46.03 | | | 100 | Pyrite |
| 14 | M17-81 | 56.26 | 32.62 | 8.25 | 1.95 | | | 0.92 | | 100 | Pitchblende |
| 15 | M17-61 | 44.08 | 32.35 | 11.88 | 1.75 | | 2.65 | 5.32 | 1.97 | 100 | coffinite |
| 16 | M17-61 | | | 0.87 | | 52.59 | 46.54 | | | 100 | Pyrite |
| 17 | M17-5 | | | | | 53.44 | 46.56 | | | 100 | Pyrite |
| 18 | M17-5 | | | | | 55.18 | 44.82 | | | 100 | Pyrite |

Calcite is the dominant gangue mineral of carbonate-type uranium ores, while uranium minerals and pyrite are the main metal minerals. From a macroscopic view, carbonate-type uranium ore occurs in the form of an independent vein, with the center composed of pitchblende and the edge dominated by calcite or alternating calcite and pitchblende, thus forming a banded structure (Figure 3A). However, microscopic studies show that pitchblende is commonly produced between or wrapped by calcite minerals (Figure 3B,C) with clear and smooth boundaries between the two mineral species. Furthermore, well-crystallized pyrite encircled by calcite minerals can sometimes be observed (Figure 3D). Pitchblende, pyrite, and calcite often show co-crystallized growth characteristics.

The gangue minerals of the fluorite uranium ore (Figure 4A) are formed during the main mineralization stage and consist mainly of small-sized fluorites and some other paragenetic gangue minerals such as quartz and calcite. Metal minerals mainly include uranium minerals (pitchblende, coffinite, brannerite) and pyrite. Under the microscope, uranium minerals mainly occur in the periphery of independent fluorite veins (Figure 4B,C) or alternately grow with fluorite minerals (Figure 4D, Table 1). Fluorite is the main gangue mineral in the central part of the vein body and occurs together with uranium minerals and pyrite. Pyrite with a good crystalline form is mainly distributed in the periphery of uranium minerals, and it spreads parallel to the vein body and occurring in the center of the vein (Figure 4B,C). SEM images show that pitchblende and pyrite either grow together between fluorite grains (Figure 4E, Table 1) or grow relatively independently between fluorite and quartz (Figure 4F, Table 1). The periphery of pitchblende is commonly enclosed by thin films of quartz (Figure 4E). The boundaries between different minerals (e.g., pitchblende and pyrite; pitchblende, fluorite, and quartz) are straight and clear and show mosaic symbiosis, which implies that uranium minerals, pyrite, and gangue minerals co-crystallized during the same stage.

The gangue minerals of silicified vein-type uranium ore (Figure 5A) are mainly microcrystalline or chalcedony quartz with different crystallinities (Figure 5B,D) that coexist with calcite or fluorite gangue minerals. Similarly, the main metal minerals are uranium minerals (pitchblende, coffinite, and brannerite) and pyrites. The microscopic and scanning electron microscopy studies showed that the center is pitchblende and the periphery is veined microcrystalline quartz (Figure 5B,C). Pitchblende and pyrite occur independently (Figure 5E,F, Table 1) or cogrow between gangue mineral-quartz grains and in straight contact with quartz crystals (Figure 5G, Table 1) (Figure 5F). In addition, the pyrites in the ore-bearing veins are euhedral and in straight contact with pitchblende or microcrystalline quartz, thus showing their cogrowth properties (Figure 5E,G).

Cataclastic altered rock-type uranium ores (also known as reddening type) are basically characterized by a deep red color with the host granites displaying cataclastic textures (or crushed gravel-like and powdery) (Figure 6A). Such cataclastic granites were strongly affected by ore-forming fluid (e.g., silicification, fluoritization, carbonation, pyritization, illitization, and chloritization), and the breccia is filled with gangue minerals such as microcrystalline quartz, fluorite, and calcite. The microscopic study showed that the deep red color of this type of ore is mainly caused by the interior altered feldspar grains or powder-like rock fragments being disseminated by the "cloud" or "star cluster" in red (Figure 6A–D). Uranium minerals not only occur as veinlet and disseminated structures in the fractures of rock breccia but also show close symbiosis with microcrystalline quartz, fluorite, and calcite gangue minerals, which are filled between breccias. It is also noteworthy that the area with red clouds is characterized by an intense distribution of fine-grained uranium minerals (Figure 6D). SEM images clearly show that pyrites with good crystals and clear boundaries are distributed in the reddening area and in the veins filled with microcrystalline quartz, fluorite, and carbonate minerals derived from ore-forming fluid (Figure 6D–H). In the absence of oxidizing traces of pyrite crystals, uranium minerals can be either wrapped in pyrite (Figure 6E), inlaid with pyrite symbiosis (Figure 6E), or grown alternately with pyrite (Figure 6G). They are also distributed independently in the microcrystalline quartz, fluorite, and calcite grains (Figure 6F) or wrapped in gangue minerals (Figure 6F).

Based on the above-mentioned studies, the spatial relationship between uranium minerals and pyrite in the Mianhuakeng deposit can be identified as follows: uranium minerals occur either in the center or periphery of vein-filling ores; alternately, they are arranged with coprecipitated gangue minerals, or they occur as veinlet and disseminated structures in cataclastic altered granite. Combined with their intact crystals, characteristics of inlaid growth, and flat boundaries between them and other gangue minerals, these observations indicate that uranium minerals and pyrites were directly produced by the same ore-forming fluid during the same stage.

## 5. Discussion of the Precipitation and Mineralization Mode of Granite-Related Uranium Deposits

Typically, the uranium precipitation mechanism in granite-related uranium deposits involves uranium transport in the form of hexavalent uranyl complexes in ore-forming fluid. The hexavalent uranyl complexes will be reduced to tetravalent uranium when they encounter reducing materials. In turn, this reduction reaction leads to an exsolution of uranium ore minerals from the ore-forming fluid.

The establishment of oxidation-reduction is based on the fact that the reductant and reduced substance (i.e., oxidant) belong to two independent systems. That is, the system containing the reductant must exist first, and then, a redox reaction occurs when a relatively oxidizing material system comes into contact with a pre-existing material system containing reductants. The result is that the oxidizing substance within the former system will be reduced by the reductant, while the reductant will also be oxidized to some extent due to the electron supply required for the reduction.

Therefore, if the traditional "reducing precipitation-mineralization" mechanism is reasonable in granite-related uranium mineralization, then the following conditions need to be met: (1) the ore-forming fluid resulting in uranium mineralization in the Mianhuakeng deposit was oxidizing; (2) the ore-forming fluid was transported from a "relatively open oxidation system" to a "pre-existing relatively reduction system"; (3) the primitive environment of ore bodies was relatively closed and reduced before mineralization; and (4) pyrite, as a reductant, was formed in an additional reduction system out of the reducing ore-forming fluid, and its formation time was much earlier than the active stage of ore-forming fluid and uranium mineralization. Furthermore, the uranium mineralization stage should be the period of pyrite consumption. In recent years, numerous scholars have carried out studies on the sources of ore-forming fluids in granite- or volcanic-related hydrothermal uranium deposits in Southeast China. A consensus has been reached that the ore-forming fluids originated from the deep crust or were closely related to crust-mantle interaction sources (or mantle materials) based on considerable geochemical and geological evidence [14,16,17,26–30].

Researchers have carried out isotopic tracer studies on the ore-forming fluids in the Mianhuakeng deposit. Based on C isotope values in ore-stage calcite ($\delta^{13}C = -9.3‰$ to $-5.3‰$), Zhang et al. [20] and Zhu et al. [31] suggested that the $\sum CO_2$ that served as a mineralizer was mainly derived from mantle degassing caused by lithosphere stretching. The results yielded by H isotopes (average $\delta D_{H2O} = -75‰$), O isotopes ($\delta^{18}O_{H2O} = 3.9‰$), and C isotopes ($\delta^{13}C = -8.4‰$ to $-5.3‰$) of ore-forming fluid and Sr isotopes of fluorite (($^{87}Sr/^{86}Sr)_i = 0.71474-0.71697$) also reflect geochemical traces of ore-forming fluid derived from mantle-derived sources [32]. The $^3He/^4He$ values analyzed for fluorite, calcite (fluid inclusions), and pyrite fluid inclusions are 0.021–1.543 Ra [22,33], which are significantly higher than the crustal $^3He/^4He$ values (0.01–0.05 Ra). This finding indicates that mantle-derived fluid was involved in the formation of ore-forming fluid in the Mianhuadeng deposit. All these suggestions are consistent with the geochemical evidence for the insufficiency of meteoric water or crust-derived fluid in forming large numbers of fluorites, calcites, and apatites found in the Mianhuakeng uranium deposit.

The source region for deep fluids is crust-mantle interaction zones or lithospheric mantle. Both regions are located in the lower parts of the lithosphere, and they are relatively closed reduction systems. Combined with the isotopic results for the Mianhuakeng deposit, a reductive fluid phase can be implied for the Mianhuakeng ore-forming fluid. Moreover, the reducing nature of ore-forming fluid in the Mianhuakeng deposit can be further deduced from reducing gas components in fluid inclusions (e.g., CO, $CH_4$, and $H_2$; [12,34]). Thus, the first condition supporting the "reducing precipitation-mineralization" mechanism is not viable in the Mianhuakeng deposit, since the ore-forming fluids are reduced rather than oxidizing.

Second, the ore-forming fluid of the Mianhuakeng deposit is thought to be related to mantle-derived fluids with the occurrence of uranium ore bodies situated in near-

vertical fault zones in the shallow crust. The structural nature of the Mianhuakeng deposit suggests ore fluid migration from deep sources (the source region likely being crust-mantle interaction zones or deep crust) to a shallow area (the current position of the ore bodies), which supports an ore-forming environment in which ore processes would have progressed from a relatively closed reduction environment to a relatively open environment. Obviously, this process contradicts the second condition that "reducing precipitation–mineralization" requires ore-forming fluids to migrate from a "relatively open oxidation system" to a "pre-existing relatively reduction system".

Third, the granites usually contain a certain amount of pyrite. To a certain extent, the granites should retain signatures of a relatively reduced system before the ore-forming fluid enters. Zhang et al. [35] and Qi et al. [36] also proposed that uranium mineralization was formed in a medium-low-temperature reduction environment. For the Mianhuakeng granites, the granite plutons are also pyrite rich. This property seems to coincide with the third condition of "reducing precipitation-mineralization". However, the ore-forming fluid at Mianhuakeng migrated and filled (into the fractures) from the deep region to the upper crust along faults or fracture zones. These faults or fractures are more likely to be open than the granite system. Although the granite itself may be reductive, the pathways for fluid migration and the vicinity (e.g., fault and fracture zone) may display a comparatively open redox nature because of infiltration from surface phreatic water.

Fourth, illitization and chloritization zones are usually developed around uranium ore bodies in the Mianhuakeng deposit, and the degree of alteration gradually decreases from the center (ore bodies) to the edge. Pyrites are highly enriched in such alteration zones, and the stronger the alteration, the higher the pyrite content. This finding indicates that these pyrites were formed during the wall rock alteration process driven by incursion of the ore-forming fluid, further implying that the pyrites are direct products of the fluid (not formed by the granite). Additionally, research from microscopic observations (see Section 4) has shown that most pyrites in uranium ores are co-crystallized with uranium minerals from the same ore-forming fluid, and they also coprecipitated at the same stage with calcite, fluorite, and microcrystalline quartz. In other words, pyrite and uranium minerals do not belong to two separate material systems. Therefore, the fourth condition for the "reducing precipitation-mineralization" theory is also not met.

It is noteworthy that the redness (reddening appearance) of the altered-rock type uranium ores and the correlation between the degree of reddening of uranium ore and the stronger mineralization of uranium ore, which are attributed to the reduction of $Fe^{3+}$ to $Fe^{2+}$ (i.e., hematization; accompanied by $U^{6+}$ oxidation), are unconvincingly regarded as important evidence for uranium "reducing precipitation-mineralization". This is because of the following reasons. (1) Physical and chemical experiments of uranium show that the process of valence change from $Fe^{2+}$ to $Fe^{3+}$ (limonitization or hematization) is actually a process of accelerating the oxidation of $U^{4+}$ and increasing the dissolution rate of uranium [5,10], which is exemplified by the significant migration of uranium in hematization zones under supergene conditions. (2) Reddening-type uranium ores occur around both sides of vein-type uranium ore bodies, and the reddening appearance in the shape of a "cloud" or "star cluster" is mainly distributed in the periphery of uranium ores in which massive co-crystallized pyrites are developed without oxidation. A large amount of pyrite coprecipitation with uranium ore is developed in the reddening area. In addition, the reducing nature of the ore-forming fluid does not match the geochemical conditions of uranium "oxidative migration-reductive precipitation", which requires an oxidizing ore-forming fluid. Therefore, the reddening mechanism of altered-rock type uranium ores needs to be further studied.

Overall, the four essential conditions required for the establishment of traditional "reducing precipitation-mineralization" in the granite-related uranium deposit are not met in the Mianhuakeng deposit. Since pyrite and pitchblende are coeval minerals with no reduction relationship between the minerals, it is unconvincing that the granite-related uranium mineralizations are mainly constrained by oxidation-reduction reactions.

The time gap between the mineralization age and the emplaced age of the host granites (>75 Ma) indicates that the granite had already cooled and consolidated when the mineralization of the Mianhuakeng deposit occurs. Deep circulation of supergene fluid rarely forms hydrothermal fluids enriched in F, C, S, Si, and P [14]. In contrast, the formation of the ore-forming fluids for the Mianhuakeng deposit is closely related to the crust-mantle source area or the involvement of mantle-derived fluids. Mantle-derived fluids are commonly characterized by high temperature, high pressure, supercritical state, and reducibility. The uranium geochemical behaviors in supercritical ore-forming fluids must be different from those in fluids with normal temperature and pressure. The basic theory of "uranium migrated in the form of $U^{6+}$ and precipitated and mineralized as $U^{4+}$" obtained from experiments that were completed under normal temperature and pressure is unavailable in granite-related uranium mineralization, which is closely associated with fluids derived from the mantle.

The solubility of tetravalent uranium under reducing conditions has been studied based on experiments corresponding to different parameters. For example, Alexander Timofeev et al. [37] suggested that uranium can be dissolved in reducing fluid in the form of the $UCl_4$ complex on the condition that the fluid is acidic brine, the temperature is higher than 100 °C, and the water vapor pressure is saturated. In addition, the concentration of the $UCl_4$ complex significantly increased as the temperature rose from 250 to 350 °C. It has also been proven that stable tetravalent uranium fluoride complexes ($UF_2^{2+}$ or $UF_4$) can be formed in solutions with $CO_2$ and F, low pH value (<5), and extremely low $fO_2$ [4]. Although $UF_4$ itself is insoluble, $UF_4$ can combine with carbonate to form a soluble complex ($[UF_2(CO_3)_3]^{4-}$; Zhang et al. [38]). Other studies have also established the solubility of pitchblende in brine solutions when the temperature is relatively high and without the supply of oxygen [39]. Liu et al. [40] placed uraninite in $KCl + Al(OH)_3$ solution under the following conditions: pH = 1.15, temperature = 400 °C, and pressure = 40 MPa, and the U contents of the solution reached 130 μg/g, indicating that the acidic solution containing alkali metals and aluminum halides is beneficial to uranium activation and migration in a high-temperature and high-pressure environment. Moreover, for the U-bearing carbonate-sulfide and carbonate-sulfide-silicate hydrothermal solutions at 200 °C and 25 MPa, lab experiments also found that the precipitation of pitchblende will be strongly enhanced because of the decrease in $CO_2$ concentration in the solutions [41]. This finding suggests that a moderate decompression boiling of uraniferous hydrothermal fluids is favorable for uranium mineralization. Accordingly, some scholars have also proposed that changes in physical and chemical conditions, such as fluid mixing, pressure reduction, and uranium concentration change, promote uranium precipitation and enrichment [42–44].

Combining the co-crystallization among uranium minerals, pyrite, and gangue minerals in the Mianhuakeng deposit with the above-mentioned experimental understandings, a practicable mechanism for granite-related uranium mineralization can be summarized as follows. The ore-forming fluid related to the source region of crust-mantle interaction or lithospheric mantle usually shows supercriticality and reducibility and is enriched in solvent components, including F, C, and Si, where uranium can be activated and migrated by combining with such solvents to form complexes in the form of $U^{4+}$. When the ore-forming fluid of the Mianhuakeng deposit migrated from a deep area through favorable structures (e.g., faults and fractures) to the shallow crust, most components became oversaturated in the fluid, and some minerals, such as calcite, fluorite, and quartz, precipitated due to drastic changes in physico-chemical conditions (i.e., pressure and temperature) caused by mixing with descending meteoritic water or the boiling cryptoexplosion effect of the fluid itself. Subsequently, uranium minerals and pyrites also precipitated. The pressure and temperature drop and pH and solubility (saturation) changes of the ore-forming fluid rather than the redox reaction caused uranium precipitation in the Mianhuakeng deposit.

## 6. Conclusions

(1) From the differing styles of uranium mineralization in the Mianhuakeng deposit, uranium ores, pyrites, and other gangue minerals (e.g., calcite, fluorite, and microcrystalline quartz) that formed during the mineralization stage all show coeval relationships indicative of co-crystallization phases from the same ore-forming fluid in the Mianhuakeng uranium deposit.

(2) Redox reactions are not crucial phases for uranium precipitation-mineralization in granite-related uranium mineralization. In contrast, the leading factors constraining the crystallization of uranium minerals and associated gangue minerals are decompression, decreasing temperature, changes in pH value, and solubility (saturation) of the ore-forming fluid.

**Author Contributions:** L.L., Z.W., and D.X., designed the project; L.L. did the original literature reviews; L.L. and Z.W. wrote and organized the paper, with a careful discussion and revision by D.X. All authors have read and agreed to the published version of the manuscript.

**Funding:** The paper was financially supported the National Natural Science Foundation of China (No. 41040019).

**Acknowledgments:** Thanks are expressed to my teachers involved in granite rocks researching in the Mianhuakeng deposit, as well as to a number of anonymous reviewers from which this article has benefited.

**Conflicts of Interest:** The authors declare no conflict of interest.

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
