# Peer review of "Relationship between Uranium Minerals and Pyrite and Its Genetic Significance in the Mianhuakeng Deposit, Northern Guangdong Province"

_minerals, doi:10.3390/min11010073_

Round 1

Reviewer 1 Report

The manuscript by Li et al. presents a petrological case study of the Mianhuakenga granite-hosted uranium deposit, south China, that may be of interest to readers of Minerals. However, the manuscript would require major revisions before it could be considered for publication in Minerals.

The main issue that would need to be addressed first and foremost are the pervasive language and grammar issues. This manuscript is one of the worst offenders in this respect that I have ever come across. As such, a complete re-write would be required before a technical review could even be attempted. This re-write should be done with the help of a professional geoscience language editor (e.g., Elsevier Language Services or similar) or a native English speaker familiar with the authors' work.

Additional concerns include the following:

  • The abstract is not stand-alone in that it fails to fully explain what your paper is about and what results you obtained from your study.
  • There is a distinct lack of referencing throughout the manuscript, in particular of relevant international papers.
  • The geology section is very short (3 paragraphs) and fails to adequately set the scene. The provided geology map lacks detail and only shows the district geology. A more detailed, deposit-scale geology map and sections are required to better illustrate the nature of the deposit. A time-space-event chart is required that clearly illustrates the main magmatic, deformation, metamorphic, hydrothermal and mineralization events recorded in the district
  • The orebody is poorly described with the description again limited to 3 short paragraphs.
  • The ore and gangue mineral paragenesis is poorly described. A paragenetic table is missing but required for a better understanding of the paragenetic sequence of the ore and gangue minerals.
  • The relative timing of some of the ore and gangue minerals has not been established beyond doubt. For example, there could be more than one generation of pyrite present in the deposit. No attempt was made by the authors to investigate such a possibility.
  • Figures 1a, b and c clearly illustrate that pitchblende postdates at least some of the calcite. In Figure 1a, the calcite gangue is cut by a massive pitchblende vein. In Figure 1b, pitchblende precipitated along and fills void in between calcite crystal boundaries (some of this are dead straight indicating calcite crystals were fully developed prior to pitchblende deposition). In Figure 1c, pitchblende also fills voids in betrwen and encloses idiomorphic calcite crystals. These types of relationships must be investigated and explained in detail.
  • No sample descriptions and locations have been provided. The authors must provide a table summarizing the main characteristics of the samples and their locations. In addition, the locations must be plotted on maps that clearly indicate the environment represented be these samples - i.e., distal versus proximal, gangue versus ore, etc.

Reviewer 2 Report

The paper is interesting, however, description of altered granitic rocks it is not very good. Reddening is product of highly evolved hematitization. More better could be described origin of chlorites, inclusive their chemical composition. I am not sure, that all SEM pictures of uraninite are really uraninites (e.g. chart 3 F). I thing that in this case uraninite is associated with coffinite. Also published analyses of uraninite could be more better, inclusive their content of Y, Pb and P.

Round 2

Reviewer 1 Report

Dear authors,

The manuscript has improved but the language and grammar issues remain. I find it difficult to further evaluate the manuscript in its current state as I am still struggling with this grammar and language issues. In my opinion, these would have to be fixed before anything else can be done. Best to use a professional language editor who is familiar with technical geoscience jargon.

Author Response

Thank you very much for your valuable advice. I have carefully checked the language and grammar problems in the article again.The modifications have been highlighted in blue.

Reviewer 2 Report

New manuscript version is OK.

Author Response

Thank you very much for your valuable advice to make my article more perfect.